

# Rituximab increases the risk of hepatitis B virus reactivation in non-Hodgkin lymphoma patients who are hepatitis B surface antigen-positive or have resolved hepatitis B virus infection in a real-world setting: a retrospective study

Yu-Fen Tsai[1,2], Ching-I Yang[1], Jeng-Shiun Du[1], Ming-Hui Lin[1], Shih-Hao Tang[1], Hui-Ching Wang[1], Shih-Feng Cho[1], Yi-Chang Liu[1,3,4], Yu-Chieh Su[1], Chia-Yen Dai[3] and Hui-Hua Hsiao[1,3,4]

[1] Division of Hematology and Oncology, Department of Internal Medicine, Kaohsiung Medical University Hospital, Kaohsiung Medical University, Kaohsiung, Taiwan
[2] Graduate Institute of Medicine, College of Medicine, Kaohsiung Medical University, Kaohsiung, Taiwan
[3] Faculty of Medicine, College of Medicine, Kaohsiung Medical University, Kaohsiung, Taiwan
[4] Department of Laboratory Medicine, Kaohsiung Medical University Hospital, Kaohsiung, Taiwan

Corresponding author
Hui-Hua Hsiao,
huhuhs@cc.kmu.edu.tw

## ABSTRACT

**Background:** Hepatitis B virus (HBV) reactivation with a hepatitis flare is a common complication in lymphoma patients treated with immunotherapy and/or chemotherapy. Anti-HBV prophylaxis is suggested for non-Hodgkin lymphoma (NHL) patients undergoing rituximab therapy, even those with resolved HBV infection. Since anti-HBV prophylaxis for patients with resolved HBV infection is not covered by national health insurance in Taiwan, a proportion of these patients receive no prophylaxis. In addition, late HBV reactivation has emerged as a new issue in recent reports, and no consensus has been reached for the optimal duration of antiviral prophylaxis. Thus, the aim of our study was to investigate the incidence and outcomes of HBV reactivation in NHL patients in a real-world setting and to study the frequency of late HBV reactivation.

**Materials:** Non-Hodgkin lymphoma patients who received rituximab and/or chemotherapy at our institute between January 2011 and December 2015 and who were hepatitis B surface antigen (HBsAg)- or hepatitis B core antibody (HBcAb)-positive were reviewed retrospectively.

**Results:** A total of 388 patients were screened between January 2011 and December 2015. In total, 196 patients were excluded because HBsAg was not assessed, HBcAb was negative or not assessed, or they were not treated with immunosuppressive therapy. Finally, the retrospective study included 62 HBsAg-positive NHL patients and 130 NHL patients with resolved HBV infection (HBsAg-negative and HBcAb-positive). During a median 30.5-month follow-up period, seven patients experienced HBV reactivation, five of whom had a hepatitis flare. The incidence of HBV reactivation did not significantly differ between the HBsAg-positive patients and the resolved HBV infection population without anti-HBV prophylaxis

(4.8% vs. 3.1%, *P* = 0.683). All patients with HBV reactivation were exposed to rituximab. Notably, late HBV reactivation was not uncommon (two of seven patients with HBV reactivation events, 28.6%). Hepatitis B virus reactivation did not influence the patients' overall survival. An age ≥65 years and an advanced disease stage were independent risk factors for poorer overall survival.

**Conclusion:** The incidence of HBV reactivation was similar between the HBsAg-positive patients with antiviral prophylaxis and the resolved HBV infection population without anti-HBV prophylaxis. All HBV reactivation events occurred in NHL patients exposed to rituximab. Late reactivation was not uncommon. The duration of regular liver function monitoring for more than 1 year after immunosuppressive therapy or after withdrawal of prophylactic antiviral therapy should be prolonged. Determining the exact optimal duration of anti-HBV prophylaxis is warranted in a future prospective study for NHL patients treated with rituximab-containing therapy.

# INTRODUCTION

Hepatitis B virus (HBV) infection is a primary cause of chronic liver disease, liver cirrhosis, and hepatocellular carcinoma. Approximately 15–20% of adults in Taiwan are chronically infected with HBV (*Sung, 1984*; *Gust, 1996*), and chronic liver disease and cirrhosis was the 10th leading cause of death in 2017. Therefore, Taiwan is a hyperendemic area for HBV infection, and HBV infection is an important health issue in Taiwan.

According to the guideline from the American Association for the Study of Liver Disease (AASLD), HBV reactivation in hepatitis B surface antigen (HBsAg)-positive patients is defined as a ≥2 log increase in HBV DNA compared to baseline or an HBV DNA level ≥1,000 IU/mL in patients with previously undetectable levels. Hepatitis B virus reactivation in patients with resolved HBV infections is defined as the reappearance of HBsAg or detectable HBV DNA. A hepatitis flare is defined as an increase in alanine aminotranferase (ALT) to greater than or equal to three times the baseline level and >100 IU/L (*Terrault et al., 2018*).

Reports of HBV reactivation after rituximab therapy started emerging a few years after FDA approval. Hepatitis B virus reactivation and hepatitis flares are well-known complications in cancer patients receiving chemotherapy. The combination regimen of rituximab and cytotoxic chemotherapy has also been found to increase the risk of hepatitis B reactivation, even in patients in whom HBV infection is resolved (i.e., HBsAg-negative but hepatitis B core antibody (HBcAb)-positive) (*Dong et al., 2013*; *Evens et al., 2011*; *Yeo et al., 2009*). Due to the high prevalence rate of chronic HBV infection in Taiwan, HBV reactivation-related complications are a great concern in lymphoma patients receiving rituximab and chemotherapy.

The recently published guidelines from the European Association for the Study of the Liver (EASL) and the AASLD both recommend that patients undergoing rituximab

therapy receive anti-HBV prophylaxis (*Lampertico et al., 2017*; *Terrault et al., 2018*). However, the recommended duration of anti-HBV prophylaxis is not consistent. Whereas the AASLD suggests that anti-HBV prophylaxis should continue for at least 6 months (or for at least 12 months for patients receiving rituximab) after the completion of immunosuppressive therapy, the EASL suggests that prophylaxis should continue for at least 12 months (or for at least 18 months for rituximab-based therapy). Moreover, late HBV reactivation (i.e., reactivation more than 12 months after completing immunosuppressive therapy (*Nakaya et al., 2016*) or after withdrawal of prophylactic antiviral therapy (*Liu et al., 2016*)) has been reported recently. However, the optimal duration of anti-HBV prophylaxis has not been determined. Furthermore, because anti-HBV prophylaxis for patients with resolved HBV infection is not covered by national health insurance in Taiwan, prophylactic anti-HBV therapy is not prescribed routinely to these patients. Therefore, we performed a retrospective study to understand the reality of HBV reactivation in patients with resolved HBV infection and to investigate the incidence and outcomes of HBV reactivation and late HBV reactivation in HBsAg-positive non-Hodgkin lymphoma (NHL) patients and in NHL patients with resolved HBV infections who were being treated with rituximab and/or chemotherapy.

## MATERIALS AND METHODS

### Patients and study design

We performed a retrospective observational study of patients newly diagnosed with NHL at Kaohsiung Medical University Hospital between January 2011 and December 2015. The inclusion criteria were as follows: (1) age 20 years or older, (2) received at least one cycle of rituximab and/or chemotherapy, and (3) were initially either HBsAg- or HBcAb-positive. The collected data included demographics, histological subtypes, treatment choice, hepatitis virus serology, HBV DNA level, and patient outcome. The baseline hepatitis virus serology and the ALT and HBV DNA level were tested during the time interval from the diagnosis of lymphoma to the first treatment. Hepatitis B surface antigen-positive patients took their antiviral prophylaxis until 6 months after cessation of chemotherapy, which was covered by national health insurance. Hepatitis B surface antigen-positive patients received regular follow-up evaluations of their hepatitis virus serology and HBV DNA levels every 3 months, whereas patients with resolved HBV infection only received hepatitis virus serological testing when the ALT level became abnormal. Overall survival was defined as the length of time from the date of diagnosis to the last date of follow-up or death. All patients were followed up until February 2017.

Informed consent was not needed because the study was a retrospective chart review. The study was approved by the Institutional Review Board of Kaohsiung Medical University Hospital, Kaohsiung, Taiwan, in 2016 (*KMUHIRB-E(I)-20160009*).

### Statistical analysis

Patient characteristics and factors contributing to HBV reactivation were analyzed using descriptive statistics and are presented as frequencies, percentages, and medians. Student's *t*-test or nonparametric statistics were utilized to test for statistically significant

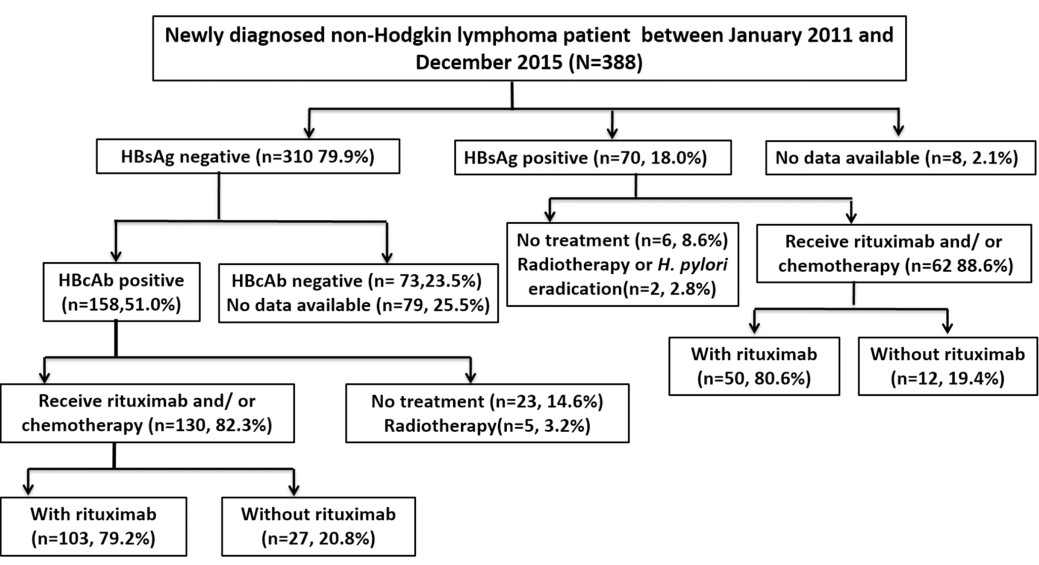

**Figure 1  Flowchart of patient selection.** HBsAg, hepatitis B surface antigen; HBcAb, hepatitis B core antibody; *H. pylori, Helicobacter pylori.*               

differences in continuous variables, whereas the chi-square or Fisher's exact test was used for categorical variables. Overall survival was estimated using the Kaplan–Meier method. Variables that were statistically significant ($P < 0.05$) in the univariate analysis of survival were subsequently subjected to multivariate analysis using a Cox regression model. A two-tailed $P$-value $< 0.05$ was considered statistically significant.

## RESULTS

### Patient characteristics

A total of 388 patients were diagnosed with NHL between January 2011 and December 2015. Seventy patients were positive for HBsAg, and 310 were negative. Among the HBsAg-negative patients, 158 were positive for HBcAb, 130 of whom received at least one cycle of rituximab or chemotherapy and thus were included in the study. Among the HBsAg-positive patients, 62 received at least one cycle of rituximab or chemotherapy and were also included. A flowchart of the patient selection process is shown in Fig. 1.

Among all included patients, 54.7% were male, and the median age at the time of diagnosis was 62 years (range, 27–90 years). The median length of follow-up was 30.5 months (range 0.8–73.9 months). The baseline patient characteristics are summarized in Table 1. The HBsAg-positive group exhibited a lower proportion of patients who were positive for HBsAb than the HBsAg-negative/HBcAb-positive group (i.e., the resolved HBV infection) (14.5% vs. 72.3%, $P < 0.001$) and was the only group to receive prophylactic antiviral therapy (98.4% vs. 0%, $P < 0.001$). No differences were found in the age, sex, lymphoma subtype, stage, rituximab use, and HBV reactivation rate between these two groups.

### Incidence of HBV reactivation and HBV-related hepatitis flares

Overall, seven patients developed HBV reactivation with an incidence rate of 3.6%, and five patients experienced an HBV-related hepatitis flare. Three of the seven patients with

**Table 1 Baseline characteristics in non-Hodgkin lymphoma patients who received rituximab and/or chemotherapy.**

| | All patients (n = 192) | HBsAg-positive group (n = 62) | HBsAg-negative/ HBcAb-positive group (n = 130) | P-value |
|---|---|---|---|---|
| Sex | | | | 0.977 |
| Male | 105 (54.7) | 34 (54.8) | 71 (54.6) | |
| Female | 87 (45.3) | 28 (45.2) | 59 (45.4) | |
| Age (years) (median, range) | 62 (27–90) | 56 (27–83) | 63 (28–90) | 0.885 |
| HBsAb | | | | 0.000 |
| Positive | 103 (53.6) | 9 (14.5) | 94 (72.3) | |
| Negative | 75 (39.1) | 49 (79.0) | 26 (20.0) | |
| Data unavailable | 14 (7.3) | 4 (6.5) | 10 (7.7) | |
| Lymphoma subtype | | | | 0.494 |
| DLBCL | 118 (61.5) | 36 (58.1) | 82 (63.1) | |
| FL | 19 (9.9) | 8 (12.9) | 11 (8.5) | |
| Marginal zone lymphoma | 21 (10.9) | 7 (11.3) | 14 (10.8) | |
| T cell lymphoma | 10 (5.2) | 1 (1.6) | 9 (6.9) | |
| MALToma | 9 (4.7) | 4 (6.5) | 5 (3.8) | |
| Mantle cell lymphoma | 7 (3.6) | 2 (3.2) | 5 (3.8) | |
| Burkitt lymphoma | 1 (0.5) | 0 | 1 (0.8) | |
| Other B cell lymphoma[a] | 7 (3.6) | 4 (6.5) | 3 (2.3) | |
| Stage | | | | 0.194 |
| I | 31 (16.1) | 13 (21.0) | 18 (13.8) | |
| II | 32 (16.7) | 6 (9.7) | 26 (20.0) | |
| III | 42 (21.9) | 12 (19.4) | 30 (23.1) | |
| IV | 87 (45.3) | 31 (50.0) | 56 (43.1) | |
| With rituximab therapy | 153 (79.7) | 50 (80.6) | 103 (79.2) | 0.820 |
| Prophylaxis with anti-HBV agents | 61 (31.8) | 61 (98.4) | 0 (0) | 0.000 |
| HBV reactivation | 7 (3.6) | 3 (4.8) | 4 (3.1) | 0.683 |

Notes:

DLBCL, diffuse large B cell lymphoma; FL, follicular lymphoma; MALT, mucosa associated lymphoid tissue; HBV, hepatitis B virus; HBsAg, hepatitis B surface antigen; HBcAb, hepatitis B core antibody; HBsAb, hepatitis B surface antibody.

[a] Other B cell lymphomas: one B cell lymphoma, one large B cell lymphoma, one B-cell lymphoma, unclassifiable with features intermediate between diffuse large B-cell lymphoma and classical Burkitt's lymphoma, one lymphoplasmacytic lymphoma, one plasmablastic lymphoma, and two small lymphocytic lymphoma.

HBV reactivation were HBsAg-positive, and four had a resolved HBV infection at the initial diagnosis. The incidence rates of HBV reactivation in the HBsAg-positive and resolved HBV infection groups were 4.8% and 3.1%, respectively, which were not significantly different.

## Factors associated with HBV reactivation

The factors attributed to HBV reactivation are compared in Table 2. Most patients experiencing HBV reactivation were older (≥60 years), had advanced-stage disease, and were predominantly male. All patients who experienced HBV reactivation received rituximab-containing treatment.

**Table 2 Factors influencing HBV reactivation in NHL patients under rituximab and/or chemotherapy treatment.**

| Factors | All patients | | | | HBsAg positive | | | | HBsAg negative/HBcAb positive (resolved HBV infection) | | | |
|---|---|---|---|---|---|---|---|---|---|---|---|---|
| | HBV reactivation | | No HBV reactivation | | HBV reactivation | | No HBV reactivation | | HBV reactivation | | No HBV reactivation | |
| | n = 7 | % | n = 185 | % | n = 3 | % | n = 59 | % | n = 4 | % | n = 126 | % |
| Age | | | | | | | | | | | | |
| <60 | 2 | 28.6 | 82 | 44.3 | 1 | 33.3 | 35 | 59.3 | 1 | 25.0 | 47 | 37.3 |
| ≥60 | 5 | 71.4 | 103 | 55.7 | 2 | 66.7 | 24 | 40.7 | 3 | 75.0 | 79 | 62.7 |
| Sex | | | | | | | | | | | | |
| Male | 5 | 71.4 | 100 | 54.1 | 2 | 66.7 | 32 | 54.2 | 3 | 75.0 | 68 | 54.0 |
| Female | 2 | 28.6 | 85 | 45.9 | 1 | 33.3 | 27 | 45.8 | 1 | 25.0 | 58 | 46.0 |
| Ann Arbor stage | | | | | | | | | | | | |
| I–II | 1 | 14.3 | 62 | 33.5 | 0 | 0 | 19 | 32.2 | 1 | 25.0 | 43 | 34.1 |
| III–IV | 6 | 85.7 | 123 | 66.5 | 3 | 100 | 40 | 67.8 | 3 | 75.0 | 83 | 65.9 |
| HBsAb | | | | | | | | | | | | |
| – | 2 | 28.6 | 73 | 39.5 | 1 | 33.3 | 48 | 81.4 | 1 | 25.0 | 25 | 19.8 |
| + | 4 | 57.1 | 99 | 53.5 | 1 | 33.3 | 8 | 13.5 | 3 | 75.0 | 91 | 72.2 |
| Unavailable | 1 | 14.3 | 13 | 7.0 | 1 | 33.3 | 3 | 5.1 | – | – | 10 | 7.9 |
| Rituximab-containing | | | | | | | | | | | | |
| – | 0 | 0 | 39 | 21.1 | 0 | 0 | 12 | 20.3 | 0 | 0 | 27 | 21.4 |
| + | 7 | 100 | 146 | 78.9 | 3 | 100 | 47 | 79.7 | 4 | 100 | 99 | 78.6 |

Note:
HBV, hepatitis B virus; NHL, non-Hodgkin lymphoma; HBsAg, hepatitis B surface antigen; HBcAb, hepatitis B core antibody; HBsAb, hepatitis B surface antibody.

## HBV reactivation and HBV-related hepatitis flares in the HBsAg-positive patients

Three of the 62 HBsAg-positive patients developed HBV reactivation. The details for these three patients are shown in Table 3. Two of these patients were male. All patients were in the advanced stage four, received multiple cycles of rituximab-containing treatment, and were receiving a different antiviral agent than that prescribed for their previous prophylaxis. Two patients developed a hepatitis flare. Patient number one received entecavir for 7.9 months, and their hepatitis was resolved without a detectable HBV DNA level. Patient number three received both tenofovir and entecavir but still died of fatal fulminant hepatitis attributed to HBV reactivation (although this patient also had comorbid hepatocellular carcinoma).

## Comparison of HBV reactivation rates between subgroups divided based on antiviral prophylaxis

Sixty-one of the 62 (98.4%) HBsAg-positive patients received anti-HBV prophylaxis. The anti-HBV prophylactic drugs administered to the HBsAg-positive patients included entecavir (35 of 61, 57.4%), tenofovir (13 of 61, 21.3%), lamivudine (nine of 61, 14.7%), and telbivudine (four of 61, 6.6%). A higher HBV reactivation rate (one of nine, 11.1%) was

**Table 3 Details of HBV reactivation and HBV-related hepatitis flares in the HBsAg-positive patients.**

| Patient number | Sex | Age | Diagnosis | Stage | HBsAb | Baseline HBV DNA level (KIU/ mL) | Prophylaxis Anti-HBV agents | Treatment regimen (# of courses) | Time to HBV reactivation (months) | HBV DNA level when reactivation (KIU/mL) | Hepatitis flare (ALT level IU/L) | Outcome and antiviral treatment after HBV reactivation | Survival state |
|---|---|---|---|---|---|---|---|---|---|---|---|---|---|
| 1 | Male | 62 | FL | 4 | Unknown | 0.727 | Telbivudine | R-CHOP (8) | 11.2 months after withdrawal of prophylaxis (late reactivation) | 44,700 | Yes (137) | ALT normal, HBV DNA level undetectable after 7.9 months of entecavir | Alive |
| 2 | Female | 37 | FL | 4 | Negative | 5.04 | Entecavir | R-CHOP (8) with R maintain (8) | During chemotherapy, 12.7 months after the first time of chemotherapy | 92,200 | No (28) | Last HBV DNA level 0.618 (KIU/mL) after 18.6 months of tenofovir | Alive |
| 3 | Male | 67 | DLBCL (HCC) | 4 | Positive | 27.8 | Tenofovir for 2 years then switched to Lamivudine | RCHOP (5) RESHAP (4) | 15.4 months after the last time of chemotherapy but still on lamivudine prophylaxis | 85,900 | Yes (2,050) | Tenofovir and entecavir for 1 week and die due to hepatic failure | Died of hepatic failure |

**Note:**
ALT, alanine transaminase; DLBCl, diffuse large B cell lymphoma; FL, follicular lymphoma; HCC, hepatocellular carcinoma; R-CHOP, rituximab, cyclophosphamide, doxorubicin, vincristine, and prednisolone; R-ESHAP, rituximab, etoposide, solu-medrol, cytarabine, and cisplatin; HBsAg, hepatitis B surface antigen; HBsAb, hepatitis B surface; R maintain, Rituximab maintenance.

found in the patients who used lamivudine as antiviral prophylaxis than in those who used entecavir, tenovir, or telbivudine (two of 52, 3.8%).

## HBV reactivation and HBV-related hepatitis flares in the patients with resolved HBV infections

Four of 130 patients with resolved HBV developed HBV reactivation; these patients are presented in Table 4. Three patients were diagnosed with a hepatitis flare, all of whom recovered after antiviral treatment (only one used lamivudine alone, whereas the other three patients used tenofovir or entecavir). Among these four HBV reactivated patients, three were male and had stage four disease. All patients were exposed to multiple cycles of rituximab-containing chemotherapy. One patient died of lymphoma progression not associated with a hepatitis flare.

### Late HBV reactivation

Two patients experienced late HBV reactivation. The probability of late HBV reactivation was not uncommon in the patients who experienced a reactivation event (28.6%). One patient had HBV that was initially resolved but reactivated 36.3 months after completing R-CHOP (rituximab with cyclophosphamide, doxorubicin, vincristine, and prednisolone) treatment; the other was initially HBsAg-positive and experienced HBV reactivation 11.2 months after withdrawal of prophylactic antiviral therapy.

### Overall survival of NHL patients

The 5-year overall survival rate was 71.0% based on the Kaplan–Meier analysis. The survival plot is shown in Fig. 2. We ran a Cox regression model to analyze factors influencing the patients' mortality (Table 5). The variables with a significant effect on mortality in the univariate analysis were included in the multivariate analysis. An older age (hazard ratio (HR) 1.03, 95% confidence interval (CI) [1.01–1.06], $P = 0.005$) and advanced disease stage (HR 4.21, 95% CI [1.78–9.95], $P = 0.001$) were associated with inferior survival, whereas the HBsAg status, HBV reactivation, and hepatitis flare did not influence overall survival.

## DISCUSSION

Our study provided real-world data for the incidence and outcomes of HBV reactivation and hepatitis flares in NHL patients. We found that the overall incidence of HBV reactivation was 3.6%; the incidence rates of HBV reactivation in the HBsAg-positive group (receiving prophylaxis) and the resolved HBV infection group (which did not receive prophylaxis) were 4.8% and 3.1%, respectively, which were not significantly different.

A retrospective Asia Lymphoma Study Group (ALSG) study of HBV reactivation in lymphoma patients revealed that this event occurred in 27.8% of HBsAg-positive patients and was significantly less frequent in patients receiving antiviral prophylaxis than in those not receiving it (22.9% vs. 59.1%; $P < 0.001$) (Kim et al., 2013). By comparison, we observed a significantly lower HBV reactivation rate (only 4.8%) in our HBsAg-positive patients, 98.4% of whom received prophylactic antiviral therapy. Lamivudine has proven to be a

**Table 4 Details of HBV reactivation and HBV-related hepatitis flares in the patients with resolved HBV infections.**

| Patient number | Sex | Age (years) | Diagnosis | Stage | Anti-HBs | Treatment regimen (# of cycles) | HBV reactivation time | HBV DNA (KIU/mL) | Hepatitis flare (ALT level: IU/L) | Outcome and antiviral treatment after HBV reactivation | Survival state |
|---|---|---|---|---|---|---|---|---|---|---|---|
| 1 | Female | 68 | FL | 4 | Negative | RCHOP (8) | 36.3 months after last course of R-CHOP (late reactivation) | 2.52* | Yes (537) | ALT normal, HBV DNA level undetectable after 13.9 months of lamivudine | Alive |
| 2 | Male | 50 | Marginal zone B cell lymphoma and transform to DLBCL | 4 | Positive | CHOP (8), ESHAP (6), R-ICE (5), auto-HSCT EPOCH (1) | During treatment, 56.0 months after first course of chemotherapy | 124,000 | Yes (121) | ALT normal, last HBV DNA level: 0.445 KIU/mL after 4.3 months of tenofovir | Died of lymphoma progression |
| 3 | Male | 73 | DLBCL | 2 | Positive | RCHOP (6) | 8.4 months after last course of R-CHOP | 12,200 | Yes (863) | ALT normal, HBV DNA level undetectable after entecavir and lamivudine together for 2 weeks and then only entecavir for 3.9 months | Alive |
| 4 | Male | 60 | FL | 4 | Positive | RCHOP (5) followed by R maintain (8) | During treatment, 24.5 months after first course of chemotherapy | >170,000 | No (98) | Just start tenofovir treatment and still pending the HBV DNA level data | Alive |

**Notes:**
ALT, alanine transaminase; DLBCL, diffuse large B cell lymphoma; EPOCH, etoposide, prednisolone, vincristine, cyclophosphamide, and doxorubicin; ESHAP, etoposide, solu-medrol, cytarabine, and cisplatin; FL, follicular lymphoma; HSCT, hematopoietic stem cell transplantation; (R)-CHOP, (rituximab)-cyclophosphamide, doxorubicin, vincristine, and prednisolone; ESHAP, etoposide, solu-medrol, cytarabine, and cisplatin; R-ICE, rituximab- ifosfamide, carboplatin, etoposide; HBsAg, hepatitis B surface antigen; HBcAb, hepatitis B core antibody; HBsAb, hepatitis B surface antibody; R maintain, Rituximab maintenance.
* HBV viral load data was checked after antiviral treatment.

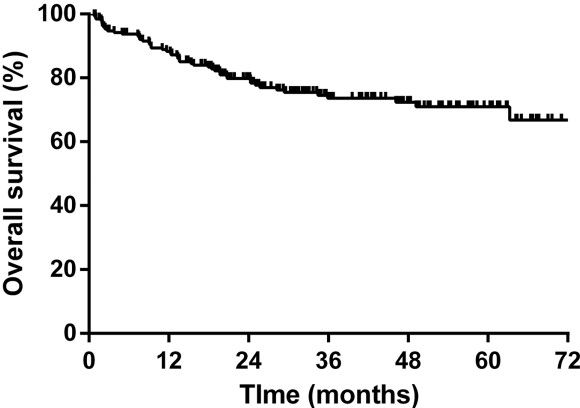

**Figure 2  Kaplan–Meier survival curve of overall survival for all NHL patients.**

**Table 5  Cox proportional hazard ratios for mortality in NHL patients.**

| Variables | Mortality | | Mortality (Adjusted) | |
|---|---|---|---|---|
| | HR [95% CI] | P-value | HR [95% CI] | P-value |
| Gender (male vs. female) | 1.59 [0.88–2.89] | 0.120 | | |
| Age | 1.03 [1.01–1.05] | 0.013* | 1.03 [1.01–1.06] | 0.005* |
| Stage (III and IV vs. I and II) | 3.86 [1.64–9.09] | 0.002* | 4.21 [1.78–9.95] | 0.001* |
| Rituximab use | 0.65 [0.34–1.22] | 0.179 | | |
| HBsAg positive | 0.69 [0.36–1.32] | 0.260 | | |
| HBV reactivation | 0.82 [0.20–3.38] | 0.779 | | |
| HBV-related hepatitis flare | 0.92 [0.22–3.82] | 0.908 | | |

Notes:
CI, confidence interval; HR, hazard ratio; NHL, non-Hodgkin lymphoma.
* P < 0.05.

useful drug to prevent HBV reactivation in patients undergoing chemotherapy for NHL (*Persico et al., 2002*). However, several studies demonstrated that patients receiving second-line entecavir as a prophylactic agent exhibited a significantly lower HBV reactivation rate than those receiving lamivudine (*Chen et al., 2013*, *2015*; *Huang et al., 2014*; *Li et al., 2011*). Furthermore, the AASLD 2018 hepatitis B guidelines recommended entecavir and tenofovir as the preferred first-line anti-HBV agents because of their higher potency and barrier to resistance (*Terrault et al., 2018*). Notably, most patients in the ALSG study used lamivudine, whereas more than 75% of the patients in our study were treated with entecavir and tenofovir for HBV prophylaxis. In addition, a higher HBV reactivation rate (11.1%) was noted in patients who used lamivudine as antiviral prophylaxis than in those who used entecavir, tenovir, or telbivudine (3.8%). This result may explain the lower HBV reactivation rate (4.8%) in our HBsAg-positive patients.

Among the population with resolved HBV infections in the ALSG study, 9.6% developed HBV reactivation (*Kim et al., 2013*), whereas the HBV reactivation rate in our study was 3.1%. Several studies have found that HBsAb negativity or a low titer is a risk factor for HBV reactivation (*Cho et al., 2016*; *Matsubara et al., 2017*; *Seto et al., 2014*;

*Yeo et al., 2009*). The higher percentage (72.3%) of HBsAb-positive patients in our resolved HBV infection group than in our HBsAg-positive group might partially explain why these patients did not experience a higher HBV reactivation rate even in the absence of anti-HBV prophylaxis. Another key factor was that we did not routinely monitor the HBV DNA and ALT levels in the patients with resolved HBV infections and only tested their HBV status upon detecting abnormal liver function. This strategy may have led to underestimation of the incidence of HBV reactivation in our resolved HBV infection group.

Regarding the features of the patients who experienced HBV reactivation, receiving a rituximab-containing regimen, an older age (≥60 years), male sex, and an advanced disease stage were associated with HBV reactivation. All patients who experienced HBV reactivation received rituximab for at least five cycles; this finding was consistent with a recent meta-analysis that revealed a significantly higher rate of HBV reactivation (8.2%) in patients receiving rituximab-containing chemotherapy than in those not receiving rituximab (0.6%) (*Evens et al., 2011*). In addition, a study focused on elderly (age >65 years) NHL patients with resolved HBV infections also reported that HBV reactivation was more likely to occur after an average of five R-CHOP cycles (*Castelli et al., 2016*). These findings all implied that multiple cycles of rituximab-containing chemotherapy was the key factor for HBV reactivation.

One patient died due to HBV reactivation-induced fatal fulminant hepatitis. However, the patient also had an underlying hepatocellular carcinoma malignancy. The remaining four hepatitis flare patients resolved after antiviral treatment, and three achieved an undetectable HBV DNA level. Overall, HBV reactivation and hepatitis flares did not compromise patient survival in our study. This result was consistent with a previous study conducted in north Taiwan in which HBV reactivation was associated with a greater number of cycles (greater than or equal to six) and a prolonged duration of rituximab therapy; overall survival in that study did not differ between patients with vs. those without HBV reactivation (*Hsiao et al., 2015*). In our study, an age ≥60 years and Ann Arbor stages III–IV were two independent negative prognostic factors.

The EASL and AASLD guidelines both recommend that patients who undergo rituximab therapy should receive anti-HBV prophylaxis (*Lampertico et al., 2017*; *Terrault et al., 2018*); however, the optimal duration of such prophylaxis remains uncertain and has varied from 6 to 18 months after completion of immunosuppressive therapy. A retrospective study conducted by *Liu et al. (2016)* found that the incidence rate of HBV reactivation after withdrawal of prophylactic antiviral therapy was 21.7% and that the median time from the withdrawal of the prophylactic antiviral therapy to HBV reactivation was 2.9 months (range, 1.1–8.5). However, several studies also reported that patients developed late HBV reactivation (i.e., more than 1 year post-therapy) (*Lee et al., 2010*; *Muraishi et al., 2017*; *Nakaya et al., 2016*; *Yamada et al., 2017*). The literature review of *Yamada et al. (2017)* revealed that an advanced disease stage, lymphoid malignancies, and treatment with multiple courses of rituximab-containing therapies were associated with late HBV reactivation. In our study, late HBV reactivation accounted for 28.6% of the total HBV reactivation events and thus was not as rare as we expected.

The longest time to HBV reactivation after completion of rituximab-containing therapy was 36.3 months. Hence, these data indicate that close monitoring of the HBV DNA and ALT levels remains important even at 1 year after completing rituximab-containing therapy or withdrawal of prophylactic antiviral therapy. Identifying patients at a high risk of late HBV reactivation and determining the optimal duration of prophylaxis for high-risk populations are pivotal aims that require further study.

Although we obtained notable results, our study had some limitations. First, our study was a retrospective observational study with a small number of patients in each group. Second, the differing chemotherapy regimens and doses may have constituted confounding factors in terms of HBV reactivation. Third, the resolved HBV infection group did not receive routine follow-up (including liver function and HBV statuses). However, our study provides clinicians the real-world data for HBV reactivation and reminds them of the possibility of late HBV reactivation.

## CONCLUSIONS

The incidence of HBV reactivation is similar between HBsAg-positive patients with antiviral prophylaxis and patients with resolved HBV infections without anti-HBV prophylaxis. Rituximab exposure is a key risk factor for HBV reactivation. Late HBV reactivation is not as rare as we expected, and regular monitoring of the HBV DNA and ALT levels is important even 1 year after withdrawal of immunosuppressive therapy or prophylactic antiviral therapy. Prospective studies are required to determine the optimal duration of prophylaxis for patients at high risk of late HBV reactivation.

### Funding
This study was supported by a grant from the Kaohsiung Medical University Hospital (KMUH103-3M12). The funders had no role in study design, data collection and analysis, decision to publish, or preparation of the manuscript.

### Grant Disclosures
The following grant information was disclosed by the authors:
Kaohsiung Medical University Hospital: KMUH103-3M12.

### Competing Interests
The authors declare that they have no competing interests.

### Author Contributions
- Yu-Fen Tsai conceived and designed the experiments, performed the experiments, analyzed the data, prepared figures and/or tables, authored or reviewed drafts of the paper, approved the final draft.
- Ching-I Yang performed the experiments, analyzed the data, collected and followed up data.
- Jeng-Shiun Du prepared figures and/or tables.

- Ming-Hui Lin prepared figures and/or tables.
- Shih-Hao Tang performed the experiments.
- Hui-Ching Wang performed the experiments.
- Shih-Feng Cho contributed reagents/materials/analysis tools, authored or reviewed drafts of the paper.
- Yi-Chang Liu conceived and designed the experiments, contributed reagents/materials/ analysis tools.
- Yu-Chieh Su contributed reagents/materials/analysis tools.
- Chia-Yen Dai authored or reviewed drafts of the paper, collected and followed up data.
- Hui-Hua Hsiao conceived and designed the experiments, authored or reviewed drafts of the paper.

## Human Ethics

The following information was supplied relating to ethical approvals (i.e., approving body and any reference numbers):

This study was approved by the Institutional Review Board of Kaohsiung Medical University Hospital (KMUHIRB-E(I)-20160009).

## Data Availability

The raw data are available in the Supplemental File.

## Supplemental Information

Supplemental information for this article can be found online at http://dx.doi.org/10.7717/ peerj.7481#supplemental-information.

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
