# Peer review of "Rituximab increases the risk of hepatitis B virus reactivation in non-Hodgkin lymphoma patients who are hepatitis B surface antigen-positive or have resolved hepatitis B virus infection in a real-world setting: a retrospective study"

_PeerJ, doi:10.7717/peerj.7481_

## Round 0.1 · original submission · Major Revisions

The authors must satisfy the critiques from the reviewers. Both reviewers 3 and 4 have recommended Rejection, and Reviewer 2 has recommended Major Revisions, so clearly there is a significant amount of work required from the authors to make the article acceptable.

Reviewer 1 ·

Basic reporting

This manuscript was written clearly. But the background should be more detailed.

Experimental design

The analysis was well designed and performed.

Validity of the findings

The data of this manuscript was well stated. It is very helpful to this field.

Additional comments

The authors analyzed the incidences and outcomes of HBV reaction in NHL patients in their hospital and found that the incidences of HBV reaction in the HBsAg-positive and resolved HBV infection population were not significantly different, and the late HBV reaction was not common. They also found that old age and advanced disease stage were independent risk factors for OS. So they concluded that regular monitoring of liver function after immunosuppressive therapy or after withdrawal of prophylactic is necessary. This retrospective analysis was well designed and performed. The manuscript would be suitable to be published, but it still need more efforts to proof-read the paper.

·

Basic reporting

I commend the authors for their effort to submit a manuscript clearly written in professional, and unambiguous language. The manuscript contains sufficient scientific validity and thus suitability to join the scholarly literature.

I thank you for providing the raw data, however, your supplemental EXCEL file contains non-English characters in the column 'diagnosis date', which should be improved to ensure that an international audience can clearly understand your text.

Your introduction needs more detail. I suggest that you improve the description of the estimated prevalence of HBsAg-carrier or chronic hepatitis B among the general population and cancer patients in the real world (in Taiwan, particularly, Kaohsiung). The scope of problem is important to your paper.

Experimental design

Kaplan-Meier curves of overall survival should be given.

Validity of the findings

The result of HBV reactivation in the comparison group of 'HBsAg-negative, HBcAb-negative or -unknown' status should be reported in the results and mentioned in the Abstract. This will help the reader understand the impact of HBsAg-positive or HBcAb-positive status on the chance of reactivation.

Additional comments

Line 201, two, not 2.

Reviewer 3 ·

Basic reporting

The topic is of interest particularly to gastroenterology consultants and researchers in the field. However, the writing is not clear and many times difficult to understand or to follow what the authors want to say. The whole manuscript needs to be reviewed by an academic whose English the mother tongue or an agent who could improve the academic writing.

Experimental design

The study design has several problems- small numbers in each group (tables 3 and 4), and different chemotherapies were used, and no proper follow up of patients.

Validity of the findings

The limitations in the methods will affect the results and it is difficult to make a generalisation in the presence of several variables and the small number of patients in subgroups (tables 3 and 4).

Additional comments

Rituximab increases the risk of hepatitis B virus reactivation in non-Hodgkin lymphoma patients who are HBsAg-positive or resolved hepatitis B virus infection in a real-world setting.

ID: #35793
Journal: Peer J


Thank you for asking me to review the above-titled manuscript which I read with great interest. However, there are several problems in the article.

The topic is of interest particularly to gastroenterology consultants and researchers in the field. However, the writing is not clear and many times difficult to understand or to follow what the authors want to say. The whole manuscript needs to be reviewed by an academic whose English the mother tongue or an agent who could improve the academic writing.

Abstract: Background: State the research question, under Materials: state what exactly was reviewed for patients with NHL. Results: First line is confusing. It is not clear what do the 62 and the 130 patients represent. Were the 130 patients negative for HBsAg and positive for HbcAg? Conclusion: not consistent with the reported results. How many patients developed late reactivation. The statement about regular monitoring of liver function tests is not consistent with what was reported in the abstract.

Introduction: Is too brief (about half a page) and did not prepare the readers to the purpose of the study. What was the rationale of the study, why this study was needed, what was the problem, what do we know about this area? The authors should justify the rationale for the study, and state their research questions. First two lines: Is Yeo et al the first author to report this reactivation? What were the recommendations of the EASL, and the AASLD? State a reference for the statement “however, prophylactic anti-HBV therapy is not prescribed routinely to patients with resolved HBV infection in clinical…”

Ethical approval for the study is a must. Please state the body/hospital/university/city/country granting the ethical approval and state the number/year of the ethical approval. What was approved exactly (line 84)? The sentence gives the impression that the institute approved that informed consent for the study was not needed because the study was retrospective!

Materials and methods: Not well described. The definition part could be placed in the introduction. The flow of writing and the formatting of the paper need improvement. What does the authors mean by “the upper limit of normal 40 IU/L at our hospital or an absolute increase >100 IU/L” which one was used? The study method could be improved by comparing prophylaxis used entecavir, tenofovir vs lamivudine. The study design has several problems- small numbers in each group (tables 3 and 4), and different chemotherapies were used, and no proper follow up of patients.

The methods need significant improvement to reflect the results outlined in tables 1 to 5. There is a need for suitable subtitles, and these subtitles should be mirrored in the results section.

Results: Lines 142-144, were these the authors results, or the results from Liue et al 2016 and Nakaya et al, 2016?
The results section needs organization and proper subtitles that mirror the methods and reflect the tables (1 to 5).

Discussion: Line 161, what do the authors mean by “the study provided real-world data”? Do other studies provide unreal world data? The limitations in the study design make it difficult to carefully interpret the results and make sound conclusions. The conclusions at the end of the study are not consistent with what was mentioned as conclusions in the abstract.

References: Several important references in the English literature were ignored such as Castelli R et al, 2016; Zhou X et al, 2017; Carrier P et al, 2015; Persico M et al, 2018 and others. The reference European Association (line 272) is not correct, why there is an electronic email address (Line 265)?, the journal “Jama” should be written “JAMA”, other similar errors were noted.

Figure 1- the description given in the abstract is not showing the grouping outlines in figure 1.

Reviewer 4 ·

Basic reporting

In general, clear and professional English language is used throughout, the manuscript is well structured and well contextualised, and raw data is supplied. The figure is clear and well labelled. References are adequate.

However there are significant errors in the tables:



In Table 1, there is no p-value presented for the difference in median age between HbsAg-positive and HbsAg- negative groups.

In Table 3 and 4, units of measurements should be provided for HBV DNA.

In table 3, the inclusion of patients 2 and 4 in the column entitled “Time to HBV reactivation after treatment completion” is confusing, because they had reactivation on treatment. The authors should include them in a separate column indicating the length of time on chemotherapy before reactivation was detected (in addition to the number of cycles).

Similarly, in table 4, the inclusion of patients 6 and 7 in the column entitled “Time to HBV reactivation after withdrawal of prophylactic antiviral therapy” is confusing, because they had reactivation on treatment. The authors should include them in a separate column indicating the length of time on chemotherapy before reactivation was detected (in addition to the number of cycles).

Furthermore, in table 4, patient 7 does not have any data about the timing of his reactivation. The table just states “during lamivudine prophylaxis”, which is inadequate.

Similarly, these tables should indicate which patients met criteria for a hepatitis flare. Currently only objective ALT levels are provided, which is inadequate.

Table 5 should be presenting hazard ratios for mortality rather than survival. It currently seems to imply that an older age or stag III/IV cancer increases the risk of survival rather than increasing the risk of mortality. Similarly, in the results section of the abstract, the last sentence should state “An age ≥65 years and advanced disease 50 stage were independent risk factors for overall mortality” (or “for poorer overall survival”).

Experimental design

The research is within the scope of the journal. Data appears to have been rigorously collected and under appropriate ethical guidance. However it does not necessarily fill a knowledge gap and indeed similar data has already been presented by other larger studies, as per the discussion.

There are significant errors and omissions in the methods though:




Patients were recruited if they were initially either HBsAg- or HBcAb-positive: what is the definition of ‘initially’- is it within a short timeframe before commencing chemotherapy? If patients’ serology results were from a long time before commencing therapy, their serological status could have changed prior to commencement.

Similarly, the time of ‘baseline’ HBV DNA measurements is not clarified- is this the level on the day of first rituximab treatment?

The methods section should clarify how often HBV DNA and sAg status was checked while patients were undergoing chemotherapy, and afterwards in both sAg positive and resolved groups. It should also state how long (in median length of time) patients who were sAg positive received antivirals after the cessation of chemotherapy.

There is no data about the median length of time of follow-up of these patients after completing chemotherapy. Thus the data about their outcomes (dead or alive) as presented in tables 3 and 4 is not meaningful. Similarly, the Cox regression analysis cannot be well interpreted given there is no information about the specific time point used in the regression analysis to deem patients dead or alive.

More description should be offered about the treatment and virological outcomes after reactivation +/- hepatitis flare was detected. In those with previously resolved infection, did commencing antiviral therapy return the HBV DNA to normal, and return the ALT to normal? If so, how long did this take?

Similarly, in those with sAg+ positivity, what were the therapeutic switches of antivirals (this is presented in table 4 but not in the manuscript)- and did these switches return the HBV DNA to normal, and return the ALT to normal? If so, how long did this take?

Line 152 states the 5-year overall survival rate was 71.0%- this should be rephrased to emphasise this was an estimate reached by Kaplan-Meier analysis rather than an objective figure. Similarly, it is not stated when is the time-point to gauge 5-year overall survival: is it 5 years after commencement of chemotherapy? Or completion of chemotherapy?

Validity of the findings

There are significant omissions in the data and overly speculative conclusions are presented.

It is stated that rituximab containing regimen, older age (≥60 years), male sex, and advanced disease stage were associated with HBV reactivation. However there is no statistical analysis that presents these risk factors as statistically significant.

It is unclear whether patient number 5, who was sAg-positive and had reactivation after withdrawal of prophylactic antivirals, had an adequate length of time of prophylactic antivirals after ceasing chemotherapy. Thus it is unclear what conclusions can be drawn from his case, if there is a possibility that he was inadequately treated (e.g. if he stopped antivirals very soon within 6 months after ceasing chemotherapy). Furthermore, it is unclear as to the intervals at which his HBV DNA status and other blood tests were checked after cessation of antivirals. His reactivation detected at 11.2 months after cessation of antivirals may actually have commenced a lot earlier but maybe was not detected if blood tests were not done routinely. Without this information, it is uncertain if he can be truly classified as “late reactivation”, and thus the overall article’s conclusion that “late reactivation is not uncommon” is specious.

Patient 2 appears to have developed reactivation after 5 cycles of rituximab. However the manuscript elsewhere states that all patients with reactivation had at least 6 cycles of rituximab. Please clarify.

The authors state in the discussion that prevalent use of entecavir and tenofovir instead of lamivudine in their HbsAg positive patients could explain low HBV reactivation rates. However they offer no statistical comparison about the rates of reactivation in those using lamivudine and in those using other antivirals to back up this claim.

It is unclear why the difference in sAb status between sAg- positive and resolved groups is emphasised not only in the results section, but also in the abstract. There is no further analysis offered about the effect of sAb positivity on the risk of HBV reactivation. The authors go on in the discussion to speculate that a high proportion of sAb positive patients in their resolved HBV group could explain low rates of viral reactivation, and in the conclusion state that “the presence of HBsAb appears to be protective against HBV reactivation”. However actual statistical comparisons between their sAb positive and sAb negative cohorts should back up this claim.






The overall aims of the study are also unclear.

Initially it is stated that the aims are only to investigate the real-world occurrences of HBV
73 reactivation in HBsAg-positive and resolved HBV infection patients.

It is thus uncertain why analyses of 5-year survival rates and Cox regression models are being emphasised as they are not relevant to, and distract significantly from, the overall message of the study. Instead of such analyses, it would be highly preferable if the authors could present statistical analysis about the risk factors for HBV reactivation (as above), and more importantly the risk factors for HBV flare. After all, it is HBV flare that is truly concerning to the clinician and potentially fatal to the patient, rather than just a rise in viral DNA.

Additional comments

There are a few unclear English language statements.

Line 152 states “The median overall survival of our NHL patients was not reached”. Please clarify what this means.

The abstract background should state “a common and potentially fatal complication”. Currently the phrase appears to imply that reactivation is invariably fatal, which is not the case.

In line 64, the phrase “This combination regimen has also been found to increase the
risk of hepatitis B reactivation” is redundant as it repeats what has been said in the previous sentence.

---

## Round 0.2 · accepted · Accept

The authors have satisfied the critiques from the reviewers